# Stromal-Modulated Epithelial-to-Mesenchymal Transition in Cancer Cells

**DOI:** 10.3390/biom13111604

**Published:** 2023-11-01

**Authors:** Huda I. Atiya, Grace Gorecki, Geyon L. Garcia, Leonard G. Frisbie, Roja Baruwal, Lan Coffman

**Affiliations:** 1Division of Hematology/Oncology, Department of Medicine, Hillman Cancer Center, University of Pittsburgh, Pittsburgh, PA 15261, USA; 2Medical Scientist Training Program, School of Medicine, University of Pittsburgh, Pittsburgh, PA 15261, USA; 3Department of Integrative Systems Biology, School of Medicine, University of Pittsburgh, Pittsburgh, PA 15261, USA; 4Molecular Pharmacology Graduate Program, University of Pittsburgh, Pittsburgh, PA 15261, USA; 5Division of Gynecologic Oncology, Department of Obstetrics, Gynecology, and Reproductive Sciences, Magee Women’s Research Institute, Pittsburgh, PA15213, USA

**Keywords:** mesenchymal stem cells, direct and indirect interaction, ECM remodeling, reprograming

## Abstract

The ability of cancer cells to detach from the primary site and metastasize is the main cause of cancer- related death among all cancer types. Epithelial-to-mesenchymal transition (EMT) is the first event of the metastatic cascade, resulting in the loss of cell–cell adhesion and the acquisition of motile and stem-like phenotypes. A critical modulator of EMT in cancer cells is the stromal tumor microenvironment (TME), which can promote the acquisition of a mesenchymal phenotype through direct interaction with cancer cells or changes to the broader microenvironment. In this review, we will explore the role of stromal cells in modulating cancer cell EMT, with particular emphasis on the function of mesenchymal stromal/stem cells (MSCs) through the activation of EMT-inducing pathways, extra cellular matrix (ECM) remodeling, immune cell alteration, and metabolic rewiring.

## 1. Introduction

Epithelial-to-mesenchymal transition EMT is a series of complex processes that enables epithelial cells to acquire a mesenchymal phenotype, leading to increased cell mobility and invasiveness. During EMT, epithelial cells undergo a series of dynamic events, including the activation of transcription factors, reorganization of the cytoskeleton and surface proteins, detachment of the basement membrane, and extra cellular matrix (ECM) degradation. EMT occurs in response to several environmental cues such as hypoxia, inflammation, and oxidative stress. Based on the broader biological context in which it is occurring, EMT is classified into three subtypes: (I) implantation and organ development, (II) response to wound healing and tissue regeneration, and (III) neoplastic progression [1,2]. In this review, we will focus on EMT associated with neoplastic progression.

In carcinoma, the EMT program is mainly associated with the acquisition of a metastatic phenotype. However, several studies also demonstrate a role for EMT during tumor initiation, whereas others suggest cancer cells maintain the activation of EMT-associated signaling throughout the metastatic cascade (extravasation, circulation, and intravasation) [3,4,5,6,7]. EMT processes appear to be triggered by different signals originating either within the carcinoma cells or from the nearby tumor-associated stroma. One critical component of the cancer-associated stroma is the cancer-associated mesenchymal stem cell (CA-MSC), which is a multi-potent stroma progenitor cell. CA-MSCs have been known to support tumor progression and metastasis in multiple cancers [8,9]. In this review, we will further highlight the role of CA-MSCs in inducing the EMT program during neoplastic progression. CA-MSCs’ signaling triggers EMT in carcinoma cells through the activation of EMT-inducing pathways, ECM remodeling, immune cell modification, and metabolic rewiring.

## 2. Activation of EMT-Inducing Pathways

It has been reported that EMT-associated intracellular signaling is induced by the complex crosstalk with stroma cells within the tumor microenvironment [10]. These intracellular cues can be activated either through the direct binding of ligands on stromal cells to their cognate receptor on cancer cells or in a paracrine manner (Figure 1). Previous work by our group and others indicates CA-MSCs contribute to cancer aggressiveness through direct interaction with cancer cells, enhancing their invasiveness and metastatic capacity in ovarian cancer [8]. The direct binding of CA-MSCs to cancer cells has also been reported to induce EMT in other cancers. Using CRISPR-Cas9 genomic perturbation and RNA-Seq, one study demonstrated MSCs via integrin B1 active Wnt/B-catenin signaling in acute lymphoblastic leukemia (ALL) cells, leading ALL cells to enter a EMT state [11]. In acute myeloid leukemia (AML), the direct co-culture of AML tumor cells and mesenchymal/fibroblastic HS5 cells results in the upregulation of the vimentin level—a main EMT pathway—and increases the metastatic phenotype of AML cells [12]. Furthermore, a study showed that the presence of a stroma component enhances EMT in BEAS-2B and HBEC-3KT lung cancer cells after a single acute exposure to radiation [13].

Along with direct binding, MSC-derived growth factors and cytokines can stimulate EMT in cancer cells in a paracrine manner. In different cancer types, including breast, ovarian, gastric, prostate, and renal cancers, MSC-derived growth factors such as transforming growth factor-β (TGF-β) have a critical role in promoting EMT in cancer cells through the activation of the SMAD signaling pathway. Along with TGF-β, other MSC-derived growth factors present in the ovarian neoplastic microenvironment, including FGF, epidermal growth factor (EGF), and hepatocyte growth factor (HGF), are thought to impair cell–cell cohesion through inducing E-cadherin cleavage, which consequentially leads to EMT [14]. Furthermore, in colorectal cancer, in vitro and in vivo studies have shown the ability of MSC-conditioned media to upregulate NF-Kb signaling through the AMPK/mTOR pathway in colorectal cancer cells [15]. Activated NF-Kb suppresses the expression of cancer cell E-cadherin and induces the expression of vimentin, which consequentially promotes EMT [16]. Furthermore, umbilical cord-derived MSC-conditioned media stimulate the Wnt signaling pathway by promoting the nuclear translocation of B-catenin in cholangiocarcinoma cell lines QBC939 [17]. Furthermore, it has been reported that stroma cells, including adipocytes and MSCs, can stimulate EMT in breast cancer cell lines through the production of CCL5 and IGF1.

Additionally, MSCs can exert their paracrine signaling through MSC-derived extracellular vesicles (EVs). MicroRNAs (miRNAs)—small non-coding RNAs—are highly enriched in MSC-derived EVs. MiRNAs have recently been identified as inducers of EMT through negatively targeted genes associated with an epithelial phenotype [18]. Mir-221/222 promotes EMT in breast cancer cell lines through targeting the estrogen receptor (ESR1) and trichorhinophalangeal syndrome type 1 (TRPS1) [19]. In gastric cancer, miR-27 increases the expression level of the EMT-associated genes ZEB1, ZEB2, SLUG, and Vimentin, and decreases E-cadherin [20].

Collectively, CA-MSCs promote EMT in cancer cells through direct and indirect interactions (Table 1). However, the changes in CA-MSCs during the direct binding are still unclear. Understanding how direct CAMSC–tumor cell interaction occurs will open up new avenues to target this interaction and potentially block EMT.

## 3. ECM Remodeling

Among the many roles that MSCs have in modulating the TME, the alteration of the extracellular matrix (ECM) to promote EMT has garnered attention over the last decade (Figure 2). The ECM serves as a scaffold for cellular attachment, a guide for cellular traffic into and out of tissue compartments, and as a stimulus for epithelial cells and tumor cells to adapt and respond to changes in their local microenvironment [21,22]. Greater investigation into MSC, ECM, and epithelial crosstalk resulted in a multifactorial understanding of how ECM content, as well as structural attributes, elicit EMT [8,23,24].

Each cancer or disease model has explored nuanced MSC features that are specific to their tissue of origin. This undoubtedly contributes to the heterogeneity of MSC function; however, the few salient mechanisms that are shared between MSC models irrefutably support the role that MSCs play in modulating the ECM to promote EMT. Chief among these effector functions are the MSC’s ability to increase the production of ECM proteins and ECM-modifying enzymes in both themselves and other stromal support cells that they signal to, the conservation of these expression changes in cells that MSCs differentiate into (e.g., CAFs), as well as the induction of reciprocal changes in epithelial and tumor cells localized to the MSC microenvironments. Each of these functions embolden the MSC influence in the surrounding tissue and synergize for effective EMT. In ovarian cancer, specifically high-grade serous ovarian carcinoma (HGSOC), there is a robust collagen deposition, resulting in densely packed regions of ECM [25,26]. An elevated deposition of collagen, among other ECM components, is associated with a worsened prognosis in patients with HGSOC [27]. The predominant type of collagen present in HGSOC tumors is collagen IV [28]. Recent studies have shown that the switch to collagen IV is advantageous for the maintenance of cancer stem cells and epithelial tumor cell proliferation [8]. Notably, the presence of collagen IV in the ECM results in the upregulation of transcription factors essential for EMT, namely, SLUG and SNAIL, via the direct interaction of collagen IV with integrins α1β1 and α2β1 [29,30]. Recent evidence has emerged showing that the utilization of collagen IV to enrich cancer cell growth and EMT is not restricted to the ovary. Rather, collagen IV regulation is utilized in other cancers (e.g., hepatocellular carcinoma) and tissues (e.g., breast epithelium) for pathologic and routine tissue maintenance [29,30]. In 2020, our group demonstrated that CA-MSCs are a potent source of collagen IV following cancer education, demonstrating that MSCs directly influence EMT in the HGSOC TME. Differential collagen production in MSCs appears to be conserved across disease processes, each fine-tuned to the unique needs of the tumor cells in each tissue. MSCs in invasive breast cancer deposit a diverse set of collagens that select for different stromal subtypes such as adipocytes (COL10A1) and chondrocytes (COL8A1, COL12A1), exemplifying an MSC’s adaptability to cancer-specific needs [31]. It is important to consider that the differential regulation of the predominant collagen subtype in the TME is occurring in the background of other ECM structural protein changes that may act in concert with collagen IV in driving EMT. The clear influence that ECM structural proteins have on epithelial and tumor cell EMT has driven investigations into therapeutics targeted towards the ECM; however, for HGSOC, no advances have been made.

As mentioned previously, simply increasing ECM structural protein production is often not the case. MSCs and their derivatives (e.g., CAFs) regulate matrix metalloproteinase (MMP) production. MMPs are a class of Zn2+-dependent enzymes that break down ECM proteins, allowing for the reorganization of the ECM structure, as well as endocrine signaling. Continuing with our example of invasive breast cancer, fibroblast derivatives of MSCs upregulate the expression of MMP-2, -3, and -9. Increased MMP-9 in the background of increased ECM structural protein deposition suggests a potential feedforward mechanism for ECM remodeling and EMT. For instance, bone-marrow-derived MSCs (BM-MSCs) are known to be recruited to sites of tissue remodeling via various growth factors and chemotactic molecules. Given the BM-MSC expression profile changes that are seen following exposure to tumor cells and other stromal cells, it is plausible that after recruitment to sites of tissue remodeling, BM-MSCs could be co-opted to embolden a pro-EMT microenvironment. The recruitment of BM-MSCs to sites of tissue remodeling is one of many downstream results of TGF-β signaling, a highly studied master regulator of EMT [32].

At a more local level, MMP upregulation serves as a three-fold selective pressure for EMT in epithelial and tumor cells by changing the ECM into an environment permissive to cell extravasation and migration, inducing the expression of MMPs in other cell types, and most importantly, by directly inducing EMT [33,34,35,36,37,38]. The pro-tumorigenic functions of MMPs, combined with specialized and increased collagen deposition, provide synergistic EMT signals, enrich cancer cell stemness, and promote epithelial and tumor cell proliferation.

In addition to MMPs, lysyl oxidases are well characterized ECM-modifying enzymes capable of inducing EMT in epithelial and tumor cells directly. Lysyl oxidases, or LOX enzymes, are a family of Cu^2+^-dependent enzymes that stabilize the ECM by crosslinking ECM constituent proteins at lysine residues. Increased ECM stiffness is a common feature among a wide variety of solid cancers and alone can elicit EMT via mechanosenstation. Not only are LOX enzymes upregulated in MSCs of breast cancer patients, but MSCs are enriched in breast tissue and in circulation [39], suggesting that LOX expressing MSCs locally drive EMT and begin to promote the formation of cancer-permissive ECM in distal tissue sites [39].

A flavor of each of these regulatory changes has been observed in numerous cancer models, including pancreatic, hepatocellular, non-small cell lung, invasive breast, and ovarian cancers both in vivo and in vitro. Given the conserved, yet highly diverse, functionality of MSCs throughout the body, future therapies targeting MSCs will require a thorough understanding of MSC tissue dynamics, as well as long-distance signaling mechanisms, along with their contribution to tumor initiation.

## 4. Immune Cell Modification

Inflammation has been implicated in various stages of tumorigenesis, including initiation, promotion, and metastasis. Chronic inflammation, driven by immune cells, cytokines, and chemokines, promotes tumor initiation and progression by creating a pro-tumorigenic microenvironment. Inflammatory cells, such as macrophages, neutrophils, and lymphocytes, infiltrate the tumor site and secrete pro-inflammatory cytokines, including interleukins (ILs) and tumor necrosis factor-alpha (TNF-α). These cytokines trigger EMT in cancer cells, leading to increased motility, invasion, and intravascular dissemination [40,41,42]. Cancer cells undergoing EMT have immune-modulatory properties that facilitate immune escape and metastatic dissemination. EMT-induced changes in cancer cells result in a reduced expression of major histocompatibility complex (MHC) molecules, leading to impaired antigen presentation and reduced recognition by cytotoxic T cells. Consequently, cancer cells evade immune surveillance and develop a resistance to immunotherapies [2].

EMT-induced immune modulation affects various immune cell populations. EMT-driven cancer cells can attract tumor-associated macrophages (TAMs) and myeloid-derived suppressor cells (MDSCs) to the tumor site, contributing to tumor growth and immunosuppression. Additionally, an upregulated expression of indoleamine 2,3-dioxygenase (IDO) enzyme by cancer cells drives effector lymphocytes, mainly T cells, towards apoptosis, further impairing the immune response [43]. This enzyme has been utilized to identify cancer activity and treatment response in clinical settings [44]. B and T regulatory cells have also been linked to tumor progression [45]. T reg might increase the population of activated CD4 þ and CD8 þ T cells IDO by affecting metabolite levels, a process of immune regulation also executed by MSCs [46,47].

MSCs engage in immune modulation through a range of mechanisms. They suppress natural killer (NK) cell activation, lessening the activation and fundamental activities of dendritic cells (DC); affect B cell proliferation and functions; and promote regulatory T cell expansion [48,49,50,51]. Another mechanism used by MSCs to immunomodulate the microenvironment around them is by direct cell-to-cell contact or the release of soluble factors. Some identified soluble factors produced by MSCs that affect the immune response are cytokines, enzymes, and nitric oxide. This process is caused by the inhibition of naïve and memory T-cell responses, consequently reducing the communication between these immune cells and antigen-presenting cells, which lessens the immune adaptive response. Via intercellular adhesion molecule-1 (ICAM-1) and vascular cell adhesion molecule-1 (VCAM-1) upregulations, MSCs reduce T-cell activation and leukocyte attraction to the inflamed area [52,53].

In vitro investigations have elucidated the direct impact of mesenchymal stem cells (MSCs) on B-cells, particularly regarding their interaction with adipose tissue-derived MSCs (A-MSCs). Through cell-to-cell contact, A-MSCs have been observed to enhance the survival of quiescent B-cells and promote B-cell differentiation independently from T-cells [54]. A-MSCs exhibit a multifaceted approach to regulate B-cell behavior; they counteract Caspase 3-mediated apoptosis by upregulating vascular endothelial growth factor (VEGF) and impede B-cell proliferation by inducing G0/G1 cell cycle arrest via the activation of p38 mitogen-activated protein kinase (MAPK) pathways [55,56]. Furthermore, A-MSCs contribute to immune homeostasis by inhibiting plasma cell formation and fostering regulatory B cell (Breg cell) generation [28]. Notably, Breg cells that produce interleukin-10 (IL-10) play a pivotal role in transforming effector CD4+ T cells into Foxp3+ regulatory T cells (Treg cells) [57]. This immunomodulation extends to the context of T cell presence, as MSCs persist in restraining B-cell proliferation [58]. The suppression of B-cell proliferation is achieved through G0/G1 cell cycle arrest induction and the secretion of Blimp-1, an antigen production factor. Significantly, MSC-mediated communication involving programmed cell death protein 1 (PD-1) is instrumental in this immunosuppressive process [59].

MSCs can secrete interleukin-10 (IL-10), and in a study using a sepsis model in mice, MSCs displayed the ability to bolster IL-10 production. This effect was found to be instrumental in enhancing overall survival, as evidenced by experiments where neutralizing IL-10 actually counteracted the beneficial impact of MSCs following sepsis induction [60]. These findings underscore the potency of IL-10 in MSC-mediated immune modulation. This IL-10 guided modulation in a non-cancerous environment may indicate that in tumors, CA-MSCs might lose the capacity to produce this interleukin. Consequently, this would be favorable to increase inflammation.

Regulatory T cells (Tregs) are important figures in immune responses and tumor progression. As cancer cells undergo epithelial-mesenchymal transition (EMT), they strategically downregulate effector molecules in CD8+ T cells, which promotes the expansion of Tregs in a TGF-B dependent process. The involvement of MSC-secreted soluble factors, including CCL-18 and transforming growth factor beta 1 (TGF-β1), guide the transformation of naïve CD4+ T cells into T regs [61]. The interplay between T regs and MSCs in cancer suggests that CA-MSCs are supportive to these cells’ proliferation and action, which affects the treatment response.

In ovarian cancer, CA-MSCs have been associated with protumorogenic behavior and chemotherapy resistance [62,63]. TGF-B1 is expressed by ovarian tumor cells and CA-MSCs (R), suggesting that this could be associated with T regs transformation; TGF-B1 was also associated with driving the expression of Tgfbi in monocytes and macrophages. This process contributes to immunosuppression and to anti–PD-L1 therapy resistance. Ovarian tumors present an inverse correlation between the presence of CA-MSCs and the abundance of intratumoral CD8+ T cells [64]. These T cells are predominantly localized within the peritumoral stroma, aligning themselves with immunosuppressive myeloid cells. CA-MSCs also release CCL2 and CX3CL1, besides TGF-B1, factors related to CCR2 monocyte recruiting and promotion to a protumorigenic M2-like phenotype [62]. In a study using a syngeneic orthotopic mouse model of ovarian cancer, the anti-tumor properties of compact bone-derived MSCs (CB-MSCs) were shown to be enhanced when combined with a fusion protein labeled as VIC-008. The intricate mechanism of VIC-008 involves the activation of CD4+ and CD8+ T-cells and the suppression of Tregs within the tumor microenvironment (TME), contributing to its anti-tumor effects [65].

Understanding the intricate interplay between EMT, inflammation, and immunomodulation in cancer presents potential therapeutic opportunities. Targeting MSCs offers promising therapeutic approaches to counter immune evasion in cancer. Further research is essential to clarify the specific interactions between immune cells and MSCs in ovarian cancer. This will facilitate the development of therapies aimed at inhibiting or reducing the transformation of normal MSCs into CA-MSCs. These therapeutic interventions have the potential to restore the anti-tumor immune response, enhance the infiltration immune cells into the tumor microenvironment, and sensitize cancer cells to immune-mediated cytotoxicity. Immunotherapy has demonstrated effectiveness in treating various solid tumors, often with a lower toxicity profile compared to standard-of-care treatments. Therefore, the pursuit of innovative therapies targeting the immune system appears to be a promising approach for enhancing the prognosis and quality of life for individuals with ovarian cancer.

## 5. Metabolic Rewiring

Metabolic reprogramming is a critical hallmark of cancer and contributes to numerous disease processes such as tumorigenesis, metastasis, and therapy resistance [66,67]. As the metabolism needs to evolve throughout cancer progression, malignant cells adapt their metabolism through a variety of cell-intrinsic and -extrinsic mechanisms, engaging in a complex crosstalk with their surrounding microenvironment to meet energy demands [67,68]. Accordingly, the process of EMT (and other motility-altering phenotypes) necessitates alterations to cellular metabolism to both meet the energy requirements for motility and survive in hostile environments outside of the solid tumor [69]. This association between metabolic reprogramming and EMT has indeed been extensively demonstrated; however, less is known regarding the cause-and-effect relationship between the two processes, including the machinations of the stromal TME in the complex EMT–metabolic crosstalk [69,70,71]. In this section, we will explore our current understanding of EMT–metabolic crosstalk and how the stromal tumor microenvironment contributes to this interplay.

Perhaps the most straightforward link between EMT and metabolic reprogramming is the overlap of multiple signaling factors that can induce both. Though TGF-β signaling has a well-documented EMT-inducing role in multiple cancer types, it also acts as a metabolic modulator both in tumor cells, as well as the broader microenvironment [72,73,74,75]. Though autocrine TGF-β signaling can occur in tumor cells, the tumor-associated stroma may secrete TGF-β and drive tumor cell EMT in a paracrine manner as well. TGF-β secreted from CA-MSCs and their derivatives (such as CAFs) have been shown to drive cancer EMT and metastasis in numerous malignancies, including colorectal, pancreatic, breast, prostate, bladder, and lung cancers [76,77,78,79,80,81].

Enhanced glycolysis is closely linked to EMT programming and can be upregulated through TGF-β signaling, which increases the expression of multiple glycolytic enzymes and glucose transporters. The expression of glucose transporter 1 (GLUT1), which mediates the influx of glucose into the cytoplasm as the first steps in glycolysis, is induced by TGF-β in gastric cancer, pancreatic ductal adenocarcinoma (PDAC), glioma, and breast cancer [82,83,84,85]. The EMT markers E-cadherin and vimentin are also correlated with the expression of GLUT1, corresponding to increased cellular glucose during TGF-β1-induced EMT in breast cancer cells [84]. Similarly, the TGF-β induction of EMT in non-small cell lung cancer (NSCLC) cells leads to GLUT3 upregulation, the inhibition of which decreases glucose uptake and proliferation [86]. In addition to promoting glucose uptake, TGF-β signaling can also increase the transcription of key glycolytic enzymes hexokinase 2 (HK2), 6-phosphofructo-2-kinase/fructose-2,6-bisphosphatase 3 (PFKFB3), and pyruvate kinase M2 (PKM2) to promote glycolysis during EMT [14,16,18,19]. PFKFB3 was also found to be upregulated by TGF-β1 in Panc1 PDAC cells, which promoted glycolysis, and when silenced, inhibited TGF-β1-induced invasion through the downregulation of SNAIL expression [85].

In addition to promoting glycolysis, TGF-β signaling can alter fatty acid metabolism by upregulating fatty acid oxidation (FAO), providing increased energy production to cells undergoing EMT. Lung A549 cells treated with TGF-β1 downregulated the major lipid metabolism regulator carbohydrate-responsive element-binding protein (ChREBP) to decrease fatty acid synthesis and promote FAO during EMT induction [75]. TGF-β2 signaling was also shown to promote FAO by increasing fatty acid uptake via CD36 [73]. The TGF-β induction of FAO at the expense of de novo fatty acid synthesis during EMT has been documented in multiple studies; interestingly, this induction appears to be context-dependent, as TGF-β can inhibit FAO and promote fatty acid synthesis as well [87,88,89].

EMT-associated transcription factors (EMT-TFs) can also alter cellular metabolism independent of TGF-β signaling. SNAIL, SLUG/TWIST, and ZEB1/2 represent a core group of well-defined EMT-TFs that couple the activation of EMT to a multitude of pro-tumorigenic functions, such as stemness, survival, and metabolic reprogramming [90,91]. In general, the activation of these particular EMT-TFs influence metabolism by promoting glycolysis and downregulating glucose-related OXPHOS; however, there are exceptions [86,92,93,94,95,96]. In breast cancer MDA-MB-231 and MCF7 cell lines, SNAIL was shown to suppress glycolysis through the suppression of the glycolytic enzyme platelet isoform of the phosphofructokinase (PFKP) under oxidative stress, shifting glucose flux to generate NADPH via the pentose phosphate pathway (PPP) [92]. EMT-TFs can also influence other aspects of cellular metabolism, such the cellular lipid pool, autophagy, and ROS; though the discussion of these processes is beyond the scope of this review, they are covered in greater detail here (Table 1) [71,90,97].

Just as the activation of EMT influences cellular metabolism, changes in cellular metabolism can subsequently promote or inhibit EMT. As mentioned above, cancer cells can adjust their metabolism to exploit and adapt to their surrounding microenvironment [66,67]. Cancer cells generally exhibit increased glucose uptake compared to normal cells, and just as the induction of the EMT program enhances glycolysis, glycolysis can also facilitate EMT [98]. There is an increased expression of glucose transporters (GLUT1, GLUT5), glycolytic pathway enzymes (HKI-III, PFK1, PGI, ALDOA/B, PGK1, ENO1, PKM2, LDH, LDHC), and glycolysis-regulating enzymes (PFK2, PFKFB3, PDK1) [99,100,101,102,103,104,105] (comprehensively reviewed in [98]). Furthermore, increased lactate production, a byproduct of aerobic glycolysis, leads to the generation of an acidic microenvironment, in turn, promoting EMT [106,107,108]. Likewise, increased fatty acid uptake and FAO can also induce EMT. Treatment with free fatty acids enhanced the EMT phenotype in hepatocellular carcinoma cells through CD36-mediated FFA uptake and TGF-β/Wnt pathway activation [109]. The overexpression of carnitine palmitoyltransferase 1A (CPT1A), an essential enzyme for FAO, increases vimentin and SNAIL and decreases E-cadherin expression in gastric cancer [110]. Increased FAO can also elevate mitochondrial ROS levels, driving EMT in high-ROS cancer cells via p38 MAPK signaling [111]. Fatty acid-binding protein 12 (FABP12) amplification in prostate cancer models leads to PPARγ activation and the concurrent induction of fatty acid uptake, FAO, and EMT [112]. Acidosis, a common trait of most TMEs, promotes autocrine TGF-β2 signaling and the formation of lipid droplets, fatty acid metabolism, and partial EMT [73].

Tumor cells engage in a dynamic, bi-directional metabolic crosstalk with the surrounding stromal microenvironment, which influences disease progression and metastasis. Likewise, stromal cells within the TME can alter the tumor cell metabolism to induce EMT. CA-MSCs and their derivatives, such as CAFs and carcinoma-associated adipocytes (CAAs), influence tumor cell metabolism through either direct methods, such as the secretion of cytokines and metabolites, or through indirect means by altering the surrounding microenvironment (such as in ECM-remodeling) [70,113]. Several studies have demonstrated that both CA-MSCs and CAFs are capable of increasing glycolytic flux in tumor cells. CAF-derived IL-6 was shown to drive glycolytic flux and select for aggressive “stem-like” CD133+ pancreatic tumor cells, enhancing their malignant potential [114]. Similarly, tumor cell-derived TGF-β signaling can trigger the release of several cytokines from CAFs, such as IL-6, CXCL10, and CCL5, which, in turn, drive the upregulation of glycogen metabolism in tumor cells through phosphoglucomutase 1 (PGM1) activation [115]. Accumulated glycogen feeds into glycolysis to drive EMT and metastasis [115,116]. The secretion of collapsin response mediator protein-2 (CRMP2) by ovarian cancer-derived CAFs was shown to upregulate the HIF-1α-glycolysis signaling pathway in SKOV3 and A2780 cells and drive EMT both in vitro and in vivo [117]. Lactate, a metabolic byproduct of aerobic glycolysis, contributes to the acidification of the TME, which can induce EMT through TGF-β signaling [73,104,114]. Lactate can be derived either from tumor or stromal cells; interestingly, lactate secretion by either cell population appears to be cancer-type specific, with prostate-derived CAFs secreting lactate, which is consumed by tumor cells (the “reverse Warburg effect”), and the opposite reporting in breast, colon, pancreatic, and ovarian CAFs [68,77,118,119,120].

**Table 1 biomolecules-13-01604-t001:** Role of stromal cells in promoting EMT in different cancer types.

Stromal Affect	Type of Regulation	Targets on Cancer Cells	Role in Cancer Cells	References’ Number	Cancer Types
Integrin B1	Direct binding	Wnt/B catenin		[11]	Acute Lymphoblastic leukemia ALL
TGF-b	Secreted factor	SMAD	Promote EMT	[14]	Breast cancer Ovarian cancer Gastric cancer Prostate cancer Renal cancers
FGF	Secreted factor	E-cadherin	Promote EMT	[14]	Ovarian cancer
EGF					
HGF			Cell–cell adhesion		
Conditioned media	Secreted factors	AMPK/mTOR	Suppress the expression of E-cadherin/ increase vimentin expression	[15,16]	Colorectal cancer
		Wnt signaling	Promote nuclear translocation of B-catenin	[17]	Cholangiocarcinoma cell lines QBC939
Collagen deposition	ECM remodeling	Upregulation of SLUG and SNAIL	Direct interaction of collagen IV with integrins a1b1 and a2b1	[29,30]	Ovarian cancer Hepatocellular carcinoma
MMPs	ECM remodeling	Provide synergistic EMT signals	Enrich cancer cell stemness, and promote epithelial and tumor cell proliferation	[33,34,35,36,37,38]	Ovarian cancer Breast cancer
lysyl oxidases	Increased ECM stiffness		Stimulate EMT via mechanosenstation	[39]	Breast cancer
Lactate	Secreted	TGF-b activation	Induce EMT	[68]	Prostate cancer
TGF-b2	Secreted factor	CD36	Promote FAO by increasing fatty acid uptake	[73]	
Acidosis	TME	Promote autocrine TGF-b2 signaling	Induce partial EMT	[73]	
TGF-b1	Secreted factor	Downregulate chREBP	Decrease fatty acid synthesis and promote fatty acid oxidation during EMT	[75]	Lung A549 cells
TGF-b	Secreted factor	Upregulation of GLUT1	Increase glucose uptake	[81,82,83,84,85]	Gastric cancer Pancreatic ductal adenocarcinoma Glioma Breast cancer
TGF-b	Secreted factor	Upregulation of GLUT3	Increase glucose uptake	[86]	Non-small cell lung carcinoma
		PFKFB3	Promote glycolysis		Panc1 PDAC cells
CPT1A	Overexpression in tumor cells	Vimentin, SNAIL, E-cadherin	Increase vimentin and SNAIL expression, decrease E-cadherin expression, enhance EMT phenotype	[110]	Gastric cancer Pancreatic ductal Adenocarcinoma Glioma Breast cancer
FABP12	Amplification in tumor cells		Induce EMT via PPARy and concurrent FAO	[112]	Prostate cancer
IL6	Secreted	CD133	Drive glycolytic flux and enhance malignant potential by enriching stemness	[114]	Pancreatic cancer
IL6, CXCL10 CCL5	Secreted	PGM1 activation	Upregulate glycogen metabolism to drive EMT and metastasis	[115,116]	
CRMP2	Secreted	HIF1A	Drive EMT via upregulation of glycolysis	[117]	Ovarian cancer

## 6. Discussion

Though the role of epithelial-to-mesenchymal transition (EMT) in cancer progression has been well established, growing evidence points to the role of stromal cells within the tumor microenvironment in promoting and synergizing with the activation of EMT. EMT is a series of complex external and internal signaling that enables epithelial cancer cells to enter a mesenchymal-transition phenotype, leading to an increase in their mobility and invasiveness. Even though EMT has been described as the first event of metastasis and as a critical step in detaching tumor cells from the primary tumors, evidence has shown that EMT signaling is present during the initiation step of cancer development. Furthermore, within the past decade, studies have mentioned the critical role of maintaining EMT signaling throughout the extravasation, circulation, and intravasation steps of the metastatic cascade. Circulating tumor cells (CTCs) present a valuable model to examine EMT-associated genes such as vimentin, E-cadherin, and N-cadherin. The upregulation of EMT- associated genes enables CTCs to survive the transit microenvironment, which is imposed by anoikis and shearing stress. Moreover, traveling as a cluster of CTCs and stroma cells not only increases CTCs’ survival rate, but also their metastatic capacity. These data suggest that the presence of stroma cells may play a role in maintaining the activation of EMT-associated signaling throughout the circulation in the bloodstream. Furthermore, in ovarian cancer, where ascites fluid is considered the mainstream for ovarian cancer circulation within the abdominal cavity, our group has shown that ovarian cancer patients’ ascites contain hetero-cellular complexes of tumor cells and stroma cells, including mesenchymal stem cells. Together, these data suggest the critical role of stroma cells, especially MSCs, in stimulating EMT not only within the primary tumor microenvironment, but also throughout the metastatic processes.

## Figures and Tables

**Figure 1 biomolecules-13-01604-f001:**
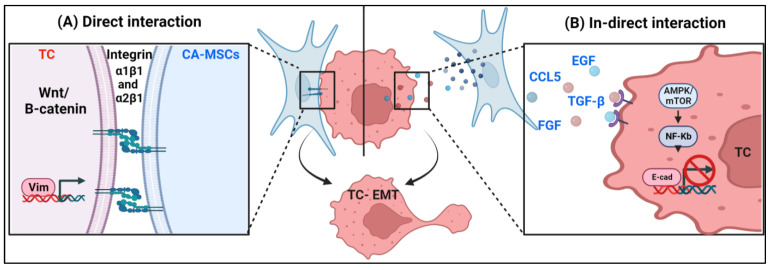
Activation of EMT-inducing pathways via cell–cell interaction. CAMSCs can activate the EMT-inducing pathway through: (**A**) direct CAMSC–tumor cell interaction; (**B**) indirect CA-MSC and tumor cell interaction via the secretion of growth factors (created with BioRender.com).

**Figure 2 biomolecules-13-01604-f002:**
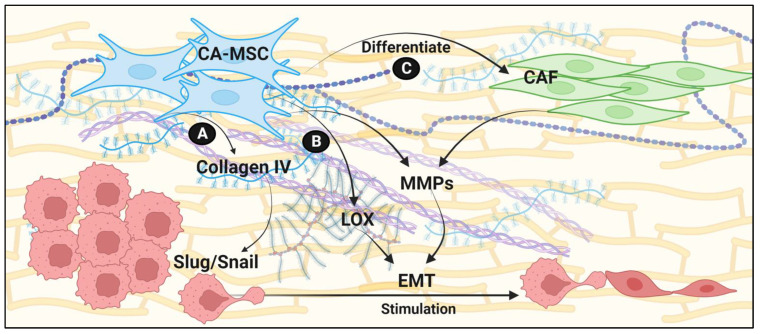
Activation of EMT-inducing pathways via extra cellular matrix (ECM) remodeling. CA-MSCs can stimulate EMT through: (**A**) collagen IV deposition; (**B**) increasing lysyl oxidase (LOX); (**C**) differentiating into cancer-associated fibroblasts (CAFs) (created with BioRender.com).

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
