# Peer review of "Stromal-Modulated Epithelial-to-Mesenchymal Transition in Cancer Cells"

_biomolecules, 2023, doi:10.3390/biom13111604_

Round 1

Reviewer 1 Report

Comments and Suggestions for Authors

Author Response

Thank you so much for your helpful suggestion, this excellent point has been taken and now we added the research gaps that we believe they are existing and our suggestion to move forward. The added sections are listed below:

Activation of EMT-inducing pathways: Collectively, CA-MSCs promote EMT in cancer cells through direct and indirect interactions. However, the changes in CA-MSCs during the direct binding is still unclear. Understanding how CAMSC-tumor cell direct interaction occur will open up new avenues to target this interaction and potentially block EMT.

Immune cell modulation: Understanding the intricate interplay between EMT, inflammation, and immunomodulation in cancer presents potential therapeutic opportunities. Targeting MSCs offers promising therapeutic approaches to counter immune evasion in cancer. Further research is essential to clarify the specific interactions between immune cells and MSCs in ovarian cancer. This will facilitate the development of therapies aimed at inhibiting or reducing the transformation of normal MSCs into CA-MSCs. These therapeutic interventions have the potential to restore the anti-tumor immune response, enhance the infiltration immune cells into the tumor microenvironment, and sensitize cancer cells to immune-mediated cytotoxicity. Immunotherapy has demonstrated effectiveness in treating various solid tumors, often with a lower toxicity profile compared to standard-of-care treatments. Therefore, the pursuit of innovative therapies targeting the immune system appears to be a promising approach for enhancing the prognosis and quality of life for individuals with ovarian cancer.

Reviewer 2 Report

Comments and Suggestions for Authors

Atiya et al. have comprehensively written the contribution of stromal tumor microenvironment in a pan-cancer perspective, with a special focus on mesenchymal stem cells. The review is well-written and nicely composed with a few amendments to be included. 

1. Although the review explains adequately, the report lacks the newly identified role of miRNAs, lncRNAs that are recently published have not been included. In particular, the role of miR-149, miR-194 in TME. I strongly recommend the authors to include these as well as other references in this manuscript.

2. It would be nice to include a comprehensive table which includes genes, type of regulation, target gene and function of stromal markers and its relevant roles in stroma and cancer as well include the appropriate references.

I look forward to the revised manuscript. 

Comments on the Quality of English Language

No comments

Author Response

1. 

This excellent point has been taken well and now we added the role of MSC-derived microRNA as a way of the paracrine signaling that MSCs exert to promote cancer cell-EMT. The description was added as follow:

Additionally, MSCs can exert their paracrine signaling through MSC-derived extracellular vesicles (EVs). MicroRNAs (miRNAs) - small non-coding RNAs- are highly enriched in MSC-derived EVs. MiRNAs has been recently identified as inducer of EMT through negatively targeted genes associated with an epithelial phenotype [18]. Mir-221/222 promote EMT in breast cancer cell lines through targeting estrogen receptor (ESR1) and trichorhinophalangeal syndrome type 1 (TRPS1) [19]. In gastric cancer, miR-27 increases the expression level of EMT associated genes ZEB1, ZEB2, Slug, and Vimentin and decreased E-cadherin [20].

2. 

Thank you so much for your helpful suggestion. Now we include a comprehensive table to summarize the ways that CA-MSCs exert to promote EMT. The table is described below:

Round 2

Reviewer 2 Report

Comments and Suggestions for Authors

All the comments raised during the first revision were properly addressed. Therefore, I recommend the manuscript to be published with no additional review.

Author Response

Thanks